EMBO
Molecular Medicine

# Household triclosan and triclocarban effects on the infant and maternal microbiome

Jessica V Ribado[1], Catherine Ley[2], Thomas D Haggerty[2], Ekaterina Tkachenko[3], Ami S Bhatt[1,3,*] [iD] & Julie Parsonnet[2,4,**] [iD]

## Abstract

In 2016, the US Food and Drug Administration banned the use of specific microbicides in some household and personal wash products due to concerns that these chemicals might induce antibiotic resistance or disrupt human microbial communities. Triclosan and triclocarban (referred to as TCs) are the most common antimicrobials in household and personal care products, but the extent to which TC exposure perturbs microbial communities in humans, particularly during infant development, was unknown. We conducted a randomized intervention of TC-containing household and personal care products during the first year following birth to characterize whether TC exposure from wash products perturbs microbial communities in mothers and their infants. Longitudinal survey of the gut microbiota using 16S ribosomal RNA amplicon sequencing showed that TC exposure from wash products did not induce global reconstruction or loss of microbial diversity of either infant or maternal gut microbiotas. Broadly antibiotic-resistant species from the phylum Proteobacteria, however, were enriched in stool samples from mothers in TC households after the introduction of triclosan-containing toothpaste. When compared by urinary triclosan level, agnostic to treatment arm, infants with higher triclosan levels also showed an enrichment of Proteobacteria species. Despite the minimal effects of TC exposure from wash products on the gut microbial community of infants and adults, detected taxonomic differences highlight the need for consumer safety testing of antimicrobial self-care products on the human microbiome and on antibiotic resistance.

**Keywords** antibiotic; microbiome; resistance; triclosan
**Subject Categories** Chromatin, Epigenetics, Genomics & Functional Genomics; Microbiology, Virology & Host Pathogen Interaction

## Introduction

Triclosan and triclocarban (TCs) are chlorinated, broad-spectrum antimicrobial chemicals found in thousands of consumer and industrial products. They are present most notably in personal wash products including toothpaste and liquid soaps (triclosan) and bar soaps (triclocarban). In 2016, a Food and Drug Administration (FDA) ruling banned the use of TCs and 17 other antimicrobial chemicals in over-the-counter wash products, driven by the concern that the use of these products contributed to antibiotic resistance and might negatively affect human health, either through endocrine disruption or modification of the human microbiota (Voelker, 2016). Notably, many other TC-containing products, such as toothpaste, fabrics, and plastic goods (including toys), were not subjected to the ban.

To date, limited data exist regarding the effects of TCs on the human microbiota (Halden, 2016). The microbes that occupy the human body in niches from the gut to the skin have diverse roles in human health, ranging from metabolic support to immunomodulation. Imbalances in these microbial communities are implicated in a wide variety of diseases (Cho & Blaser, 2012). The extent to which triclosan exposure may induce microbial perturbations has been studied in fish and rodent models with conflicting outcomes (reviewed in Yee *et al*, 2016). Triclosan exposure restructures the juvenile fish microbiome (Narrowe *et al*, 2015) but results in recoverable alterations following short-term perturbation in adult fish (Gaulke *et al*, 2016). Adolescent rats receiving oral triclosan at levels comparable to human exposures develop lower microbial diversity in the gut and more prominent changes in taxonomic composition than in adult rats (Hu *et al*, 2016). While triclocarban exposures are less studied, in pregnant rats and their offspring < 10 days old exposure leads to lowered phylogenetic diversity and revealed a dominance of the Proteobacteria phylum in the gut (Kennedy *et al*, 2016). In a small, randomized, crossover human study, TC wash product exposure did not induce major perturbations of the oral and gut microbiomes (Poole *et al*, 2016). This finding supports other studies that have shown minimal impact of triclosan on dental microbial ecology, despite slowing the progression of periodontitis (Seymour *et al*, 2017).

1 Department of Genetics, Stanford University, Stanford, CA, USA
2 Division of Infectious Diseases and Geographic Medicine, Department of Medicine, Stanford University, Stanford, CA, USA
3 Division of Hematology, Department of Medicine, Stanford University, Stanford, CA, USA
4 Division of Epidemiology, Department of Health Research and Policy, Stanford University, Stanford, CA, USA
*Corresponding author. Tel: +1 650 498 4438; E-mail: asbhatt@stanford.edu
**Corresponding author. Tel: +1 650 725 4561; E-mail: parsonnt@stanford.edu

The core microbiome of humans is established in the first few years of life (Palmer *et al*, 2007). Disruptions to the microbiota early in development by extrinsic factors, such as antibiotics, can have long-term impacts on metabolic regulation (Cox *et al*, 2014) and can delay normal microbiota maturation (Nobel *et al*, 2015). The impact of TC exposure through household and personal care products on the developing microbiota is unknown. This study leverages a nested, randomized intervention within Stanford's Outcomes Research in Kids (STORK), a prospective cohort study of healthy mothers and infants (Ley *et al*, 2016). Specifically, we provided households with commercially available wash products containing or not containing TCs (TC and nTC arms, respectively) to evaluate the relative impact of TCs on the maternal and infant gut microbiota over the first year of the infant's life.

## Results

### Study demographics

Thirty-nine of 136 households from the STORK cohort met our inclusion criteria (i.e., at least five of six expected stool samples available from the mother and infant). Complete sampling for both infants and mothers for three time points after birth was available for 26 households, and one sample was missing for 13 households. Home visits and sample collection occurred on average 74 (14–124), 200 (135–256), and 317 (241–377) days following birth (Appendix Fig S1). These days correspond to approximately 2.7, 6.6, and 10.6 months, referred to as 2, 6, and 10 months hereafter. The average age of mothers in this subset was 34 years and 46% were of Hispanic origin (Table 1).

### Randomization to TC-containing household and personal products is sufficient to increase triclosan exposure

Urinary triclosan levels were measured as a quantitative indicator of short-term triclosan exposure. Mothers in TC households had higher spot urinary triclosan levels at 6 months when compared to those in nTC households, with a median triclosan measurement of 916.10 pg/µl compared to 76 pg/µl (Mann–Whitney *U*-test $P = 5.66e-5$). With one exception, levels in infants were uniformly low, with a median of 43.00 pg/µl in TC households and 10.05 pg/µl in nTC households (Mann–Whitney *U*-test $P = 0.06$, Fig 1).

### Mother and infants have distinct microbiome compositions not driven by randomization to TC-containing products

A principal coordinate analysis (PCoA) was performed to identify variability between the taxonomic structure of the samples. PCoA showed that samples segregated primarily by age, with 31.9% of variation among samples explained by the first two principal components (Fig 2A). By 10 months of age, infant samples clustered more closely to the mothers' samples (Fig 2B). Individual samples from TC and nTC households were evenly dispersed through the axes for both infants and mothers. Taxonomic classifications suggest that samples were generally similar at the phylum level among infants and among mothers, regardless of treatment arm (Fig EV1). Maternal samples from the various time points

**Table 1.  Selected characteristics of study sample (*N* = 39 households).**

| Treatment arm | TC | nTC |
|---|---|---|
| Number of households (*N*) | 17 | 22 |
| Individuals residing in household | 4 (3–10) | 3 (2–11) |
| Maternal age (years) | 33 (26–41) | 34 (28–42) |
| Ethnicity | | |
| Hispanic | 11 (64.7%) | 7 (31.8%) |
| Non-Hispanic | 6 (35.3%) | 15 (68.2%) |
| Bathing habits | | |
| Less than daily | 5 (29.4%) | 3 (13.6%) |
| At least daily | 12 (70.6%) | 19 (86.4%) |
| Pets | | |
| Yes | 6 (35.3%) | 5 (22.7%) |
| No | 11 (64.7%) | 17 (77.3%) |
| Mother's use of cleaning products at work | | |
| Yes | 5 (29.4%) | 6 (27.3%) |
| No | 5 (29.4%) | 11 (50.0%) |
| Not applicable | 7 (41.2%) | 5 (22.3%) |
| Days after delivery for each sample collection window | | |
| 2 months | 88 (14–124) | 74 (27–122) |
| 6 months | 199 (135–256) | 199 (146–255) |
| 10 months | 330 (259–375) | 319 (241–277) |

Demographics are self-reported at the time of enrollment. Age, individuals in the household, and the sample collection windows are reported as the median and range. Sample collection times are relative to the infant birth date. Not applicable for cleaning products at work indicates the mothers were unemployed at the time, and not receiving additional TC exposure similar to mothers that do not work with cleaning products.

clustered more closely by individual than by time point (Fig EV2A), and we did not observe major variations in taxonomic structure between maternal samples based on TC exposure using PCoA (Figs EV2B). Statistical comparisons of treatment arms with permutational multivariate analyses (PERMANOVA) showed no significant association between TC exposure and microbiome composition for infants ($R^2 = 0.012$, $P = 0.17$) but did demonstrate a minor but significant association between TC exposure and microbiome composition in the mothers ($R^2 = 0.028$, $P = 0.001$). The low contribution to variance of factors known to influence microbial colonization in infants by 2 months of age, such as delivery method ($R^2 = 0.068$, $P = 0.011$) and breast feeding ($R^2 = 0.13$, $P = 0.041$) in the household (Fig EV3), is consistent with previously reported data (Chu *et al*, 2017).

### Randomization to TC-containing products does not decrease gut microbial diversity or species richness in infants or mothers

Randomization to the TC arm was not associated with decreased gut microbiota diversity for infants or mothers at any of the 2-, 6-, or 10-month visits after infant birth (Fig 3). Specifically, Shannon diversity was not decreased in infants randomized to TC-containing products (Mann–Whitney *U*-test *P*-values for 2, 6, and 10 months: 0.66, 0.84, 0.49). As expected, microbial diversity increased as the infants progressed through the first year of life ($P = 4.4e-4$)

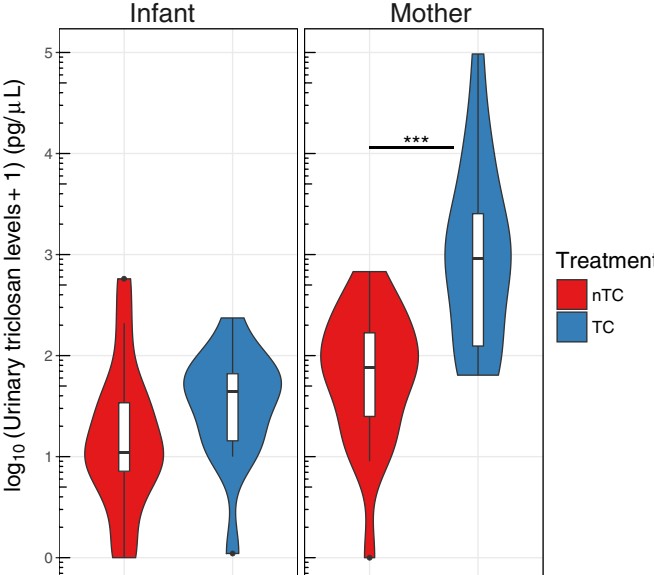

**Figure 1.  Urinary triclosan levels are elevated in TC mothers following 6 months of exposure.**

The concentrations of triclosan in picograms/microliter (pg/μl) are plotted as a log10 transformation of the absolute concentration +1, for visual clarity. The median value is represented as a black line, the interquartile range is represented by the box plot. The violin plot represents the full range of values obtained from the source data, where the width of the blue or red colored region represents the probability density of sample values at that level. The black dots represent outliers. Urinary triclosan measurements are available for 38 mothers (17 TC, 21 nTC) and 33 infants (15 TC, 18 nTC). The ranges for the violin plots are found in the source data. The $P$-value for the differences in urinary triclosan levels between treatment arms in the mothers (represented by ***) is $P = 5.66\text{e-}5$ (Mann–Whitney $U$-test).

Source data are available online for this figure.

(Yatsunenko *et al*, 2012), and this effect was not altered by randomization to TC. Diversity was not statistically significantly decreased in maternal samples randomized to TC-containing products ($P$-values for 2, 6, and 10 months: 0.73, 0.28, 0.20) but trended to a decrease in stool diversity at the 10-month visit. Microbial diversity did not correlate with urinary triclosan levels at 6 months (Infants: $R^2 = 0.024$, $P = 0.40$; Mothers: $R^2 = 0.015$, $P = 0.49$, Fig EV4). Species richness, measured by Chao1, was also not affected by TC exposure ($P$-values for 2, 6, and 10 months for infants: 0.61, 0.51, 1; mothers: 0.68, 0.49, 0.84).

**Intestinal exposure to triclosan through toothpaste, rather than wash products, is associated with Proteobacteria enrichment in TC households**

Given a minor but statistically significant difference between maternal gut compositions between the TC and nTC arms from permutation tests, we hypothesized that TC exposure affected a small proportion of taxa within the community. We identified differentially abundant taxa present in stool samples from TC vs. nTC infants and mothers by first pooling data from all three visits and separately for each of the three visits. A greater number of taxa were significantly differentially abundant in mothers vs. infants (29 in mothers, 17 in

infants; Fig 4A). Infants did not show an enrichment of specific phyla at any time point, but *Bacteroides fragilis* was persistently the most enriched species in nTC infants in combined and time-point analyses at 6 months (Fig 4A and B). Relative abundances of *B. fragilis* at this visit were not associated with delivery method (Mann–Whitney $U$-test $P = 0.85$) or breast feeding (ANOVA $P = 0.52$) as previously reported (Martin *et al*, 2016). Mothers showed a strong enrichment of Proteobacteria in the TC arm after the introduction of triclosan-containing toothpaste at the 2-month visit (toothpaste was not provided prior to the infants' births) (Fig 4C). We additionally compared differential taxa in the upper and lower tertiles of urinary triclosan levels at the 6-month visits, agnostic to treatment assignment. In the mothers, the taxon with the highest enrichment in the higher triclosan levels is a member of the Proteobacteria phylum. Infants with higher triclosan levels showed a statistically significant enrichment of the Proteobacteria phylum in their stools samples; conversely, *B. fragilis* was enriched in infants with lower triclosan levels (FDR-adjusted $P$-value < 0.1, Appendix Table S1).

Proteobacteria enrichment has been associated but is not directly correlated with increased antibiotic resistance genes in the human gut microbiota, independent of antibiotic exposure (Bengtsson-Palme *et al*, 2015). This suggests Proteobacteria species may not harbor antibiotic resistance genes in the gut microbiome, but expansion may serve as marker for increased antibiotic resistance in a community. A preliminary, unbiased quantification of community triclosan resistance in the mothers was conducted using whole shotgun sequencing of stool samples for a subset of 12 mothers in each intervention arm at 6 months. This resulted in low coverage of triclosan resistance genes with only 0.03 % of sequenced reads mapping to triclosan antibiotic resistance genes in the Comprehensive Antibiotic Resistance Database (CARD). Unsupervised clustering by triclosan resistance gene counts was not sufficient to cluster maternal samples by intervention arm, and this may have been impacted by the limited power to detect differences between groups due to low coverage (Fig EV5). Differential gene analysis from CARD showed an enrichment of one beta-lactamase antibiotic resistance gene, CfxA6, in TC households at a FDR-adjusted $P$-value < 0.1 threshold.

## Discussion

At the time of the US FDA ruling that banned 19 antimicrobials from wash products, the extent to which TC exposure perturbed microbial populations in humans, particularly during infant development, was unknown. To test the hypothesis that exposure to TC-containing wash products induces a measurable impact on the gut microbiota of adults and growing infants, we assessed the stool microbiome from mothers and infants in households that had been randomized to TC or nTC wash products during the first year of the infant's life. We observed that ongoing TC exposure from household products does not contribute to major reconstruction of either infant or adult intestinal microbiomes after approximately 10 months. TC exposure did not reduce overall gut microbial diversity in infants or mothers at any visit. However, there are some notable trends in differential taxa with potential health implications. The most enriched species in the nTC randomized infants, *B. fragilis*, has been shown to direct maturation of the immune system (Mazmanian *et al*, 2005) and produce anti-inflammatory polysaccharides (Mazmanian *et al*,

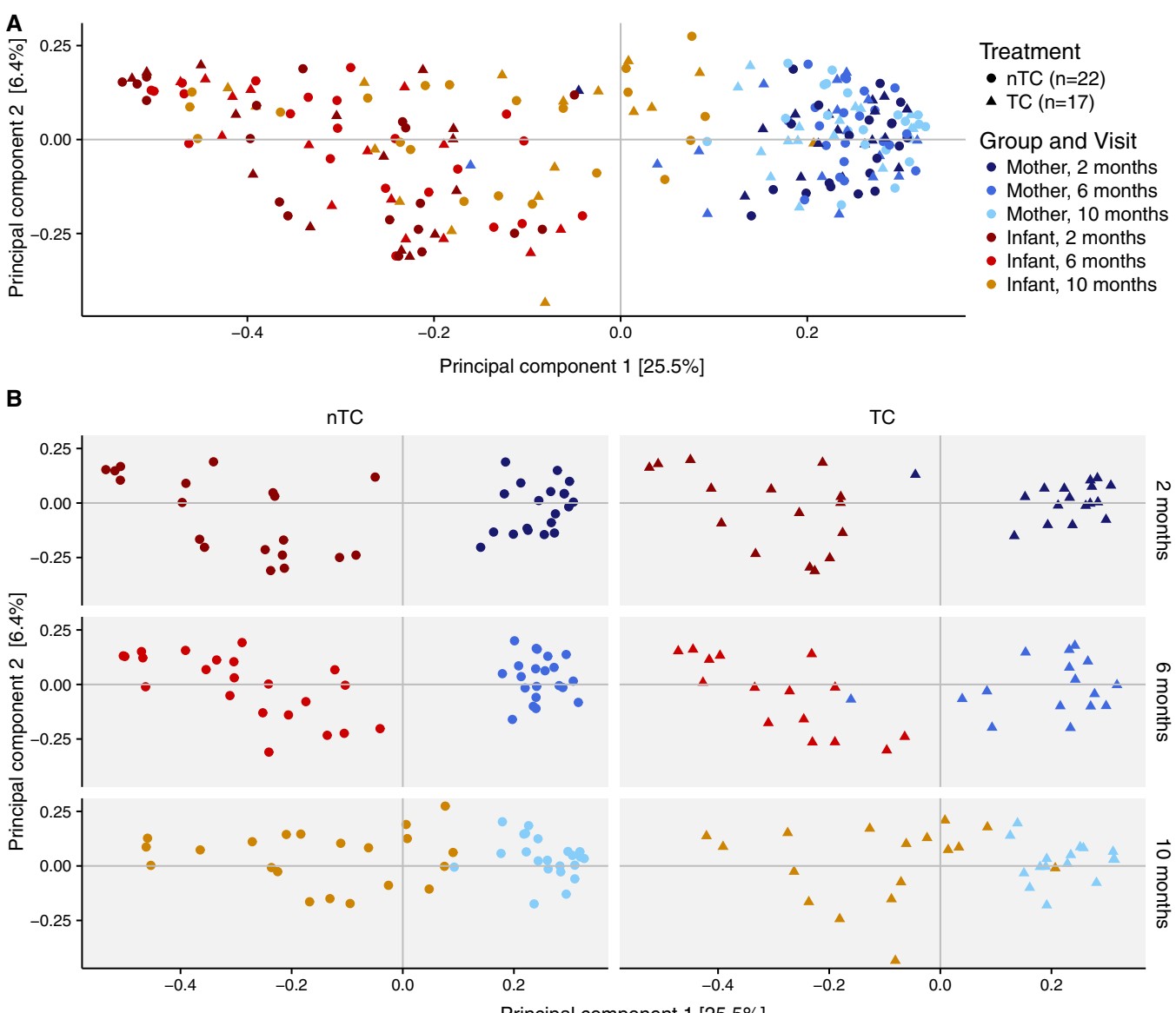

**Figure 2.  Mother and infants have distinct microbiome compositions not driven by household TC exposure.**

A   PCoA of Bray–Curtis dissimilarity for all (*n* = 221) samples shows that gut communities cluster by mothers and infants.

B   PCoA separated by time and treatment.

Source data are available online for this figure.

2008). The most enriched organisms in the TC households at the 10-month visit were *Bacteroides caccae* in infants and *Escherichia coli* in mothers. Strain-specific triclosan resistance in *E. coli* has been described (Braoudaki & Hilton, 2004) and may explain its enrichment in TC households at the late time point. Given that *B. caccae* was extensively enriched in the infants at the 10-month visit, it is possible that this organism also harbors interesting strategies for acquired or innate antimicrobial resistance.

In the TC arm of the study, mothers showed a strong enrichment of Proteobacteria. The Proteobacteria phylum contains both pathogenic and nonpathogenic species, which makes it difficult to ascribe a clinical or public health relevance to our results. While

a selective expansion of Proteobacteria is not known to cause disease, Proteobacteria expansion has been proposed as a potential diagnostic signature of dysbiosis linked to diabetes, colitis, and malnutrition (Shin *et al*, 2015). Future studies may illuminate the impact of these shifts on health-related outcomes. Proteobacteria enrichment has been associated, but is not directly correlated, with increased antibiotic resistance genes in the human gut microbiota, and perhaps can be used as a marker for community antibiotic resistance (Bengtsson-Palme *et al*, 2015). Potential antibiotic-resistant strains identified in this study may not be pathogenic, but the possibility for horizontal gene transfer with pathogenic strains between people and the environment may

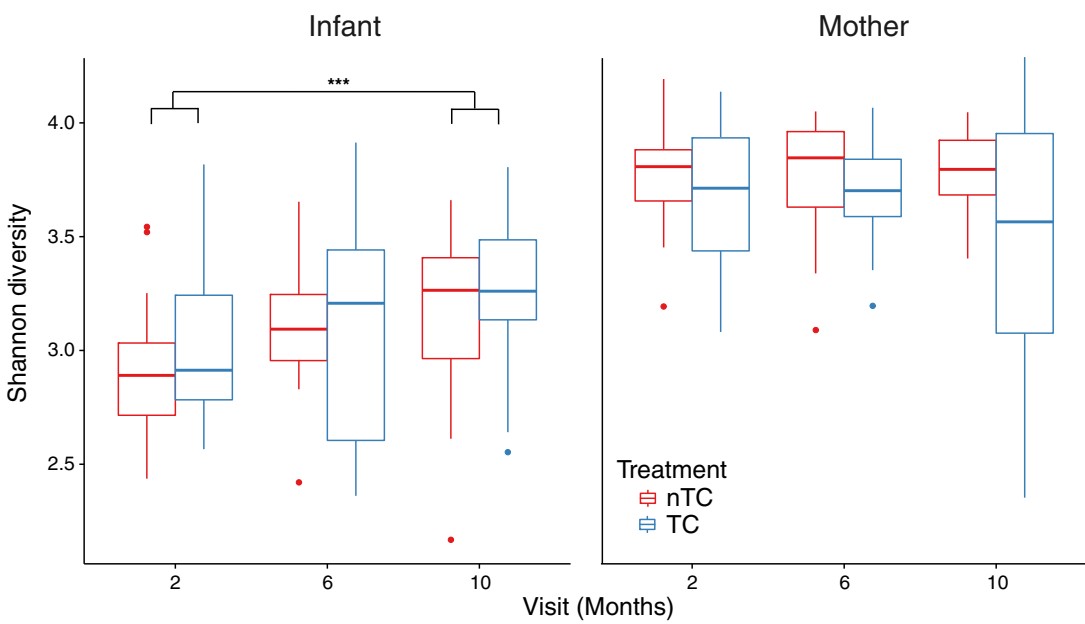

**Figure 3.  TC randomization does not decrease gut microbial diversity in infants or mothers.**
Shannon diversity measures are plotted as the interquartile range with median for each TC exposure class and time-point grouping for infants and mothers. The upper and lower whisker for box plots indicate ± 1.5× interquartile range. The points plotted beyond these whiskers are considered outliers and are plotted individually. The *P*-value represented for the increase in infant gut microbiota diversity over time is \*\*\**P* = 4.4e-4 (Mann–Whitney *U*-test).

Source data are available online for this figure.

severely exacerbate problems with population and clinical resistance. Evidence that antibiotic resistance develops in diverse bacterial taxa following prolonged triclosan exposure suggests that triclosan resistance may be mediated by specific genes (Forbes *et al*, 2016; Hartmann *et al*, 2016; Khan *et al*, 2016) and that these genes may be horizontally transferred (Ciusa *et al*, 2012). Although we only identified one significantly enriched antibiotic resistance gene in the TC-exposed mothers, point mutations in the candidate, CfxA6, have led to antibiotic resistance in the gut following antibiotic exposure (Raymond *et al*, 2016). Future *in vitro* and potentially *in vivo* studies will be required to more thoroughly characterize the impact of TCs on antibiotic resistance in the gut microbiota. The finding that Proteobacteria were enriched in the gut microbiota is consistent with observations in fish following triclosan exposure (Narrowe *et al*, 2015). The emergence of Proteobacteria was only observed after the introduction of triclosan-containing toothpaste at the 2-month visit. This suggests that the major intestinal exposure to triclosan is through toothpaste rather than wash products and that personal care products not covered by the FDA ban may play a role in the expansion of antibiotic resistance in the intestine.

One caveat of this study is that triclosan has a relatively short half-life (24 h) and urinary triclosan levels depend on recent rather than sustained exposure; thus, it is not possible to know the cumulative dose of triclosan received by the study participants by absorption through the skin or intestine. Although mothers in the TC arm had higher levels of triclosan detected in urine than those in nTC households, the median triclosan level we detected in urine of nTC households was higher than those reported in the National Health and Nutrition Examination Surveys (NHANES)

cohort from 2003 to 2012 (geometric mean concentration of 13.0 pg/µl, 95th percentile concentration of 459.0 pg/µl) (Han *et al*, 2016). In addition, the urinary triclosan levels of mothers in TC households were lower than might have been anticipated—approximately half of the levels following a 4-month randomization to household TC-containing products in a previously published crossover study (Poole *et al*, 2016). Some of the differences may be due to methodological variations in the protocols used for triclosan detection or differences in exposure due to geography and inconsistent product usage. Infants in TC households and nTC households often had low triclosan levels with no statistically significant difference observed between groups. Low levels are unsurprising, as infants are not using triclosan-containing toothpaste in the first year of life.

Exposure to other microbicides, including antibiotics administered throughout pregnancy and the first year, might also have influenced prenatal and postnatal microbiomes in ways that cannot be experimentally controlled. For infants, antibiotic use was common but similar in the two arms of the study (76.4% in the nTC and 60% in the TC arm reported receiving antibiotics). It remains possible that the dearth of taxonomic differences in the infant samples was caused by antibiotic perturbations that overwhelmed the TC effects. However, the statistically significant Proteobacteria enrichment in infants with higher urinary triclosan levels suggests that the small microbiota disruptions observed in the study were TC dose dependent.

The impact of likely low-dose but long-term (> 4 months) household product-based antimicrobial exposure on the human gut microbiome has not been previously described in either adults or infants during the critical phases of microbiota assembly early in

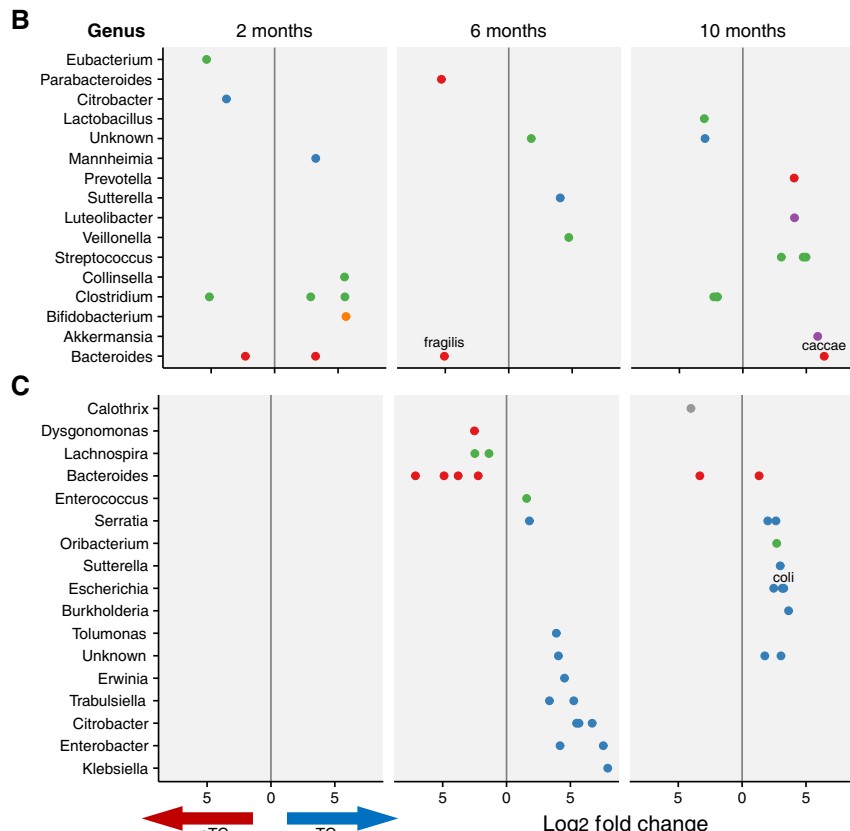

**Figure 4. Enrichment of Proteobacteria is observed in the mothers of TC households.**

A    Differentially abundant taxa between nTC and TC households. Values left of the gray line indicate an enrichment in nTC household and values to the right indicate an enrichment in TC households. Analyses are separated by mothers and infants for all samples across the three time points (FDR-adjusted *P*-value < 0.01).

B, C    Differentially abundant taxa are displayed for (B) infants and (C) mothers at per visit (FDR-adjusted *P*-value < 0.05).

Source data are available online for this figure.

life. While the impact appears to be small, we do identify specific taxa previously associated with anti-inflammatory properties that are enriched in infants of nTC households, as well as other taxa previously associated with broad-spectrum antibiotic resistance that are enriched in mothers of TC households. The measurable shift in the gut microbiome that occurs in mothers between the first and second visits of this study, which corresponds to the introduction of triclosan through toothpaste, suggests that toothpaste exposure places more selective pressure on the gut microbiome than wash products. Triclosan exposure is known to play a role in allergen and food sensitization (Hampton, 2011; Savage *et al*, 2012); topical skin application of triclosan is sufficient to induce peanut sensitivity in mice (Tobar *et al*, 2016). Given the high prevalence of TC exposure on the skin in this study, it will be interesting to study the impact of these wash products on the skin microbiota and related health outcomes. Despite the minimal effects of TC exposure from wash products on the gut microbial community of infants and adults, detected Proteobacteria enrichment highlights the need for consumer safety testing of consumer antimicrobial products on the human microbiome and antibiotic resistance.

## Materials and Methods

### Study design

Subjects in this study were recruited to participate in Stanford's Outcomes Research in Kids (STORK), a prospective cohort study of healthy mothers and infants (Ley *et al*, 2016). Briefly, pregnant mothers were enrolled in the study at approximately 20 weeks of gestation from both Lucile Packard Children's Hospital (Stanford, CA, USA) and the Tully Road Clinic of Santa Clara Valley Medical Center (San Jose, CA, USA). Enrolled mothers were additionally invited to participate in a nested, randomized intervention of TC-containing household and personal wash products to study the effects of these microbicides on illness and the development of the infant microbiome.

Participants were provided commercially available wash products (liquid and bar soap, toothpaste, dishwashing liquid) all either containing or not containing TCs. Bar soap was the only provided product that contained triclocarban in addition to triclosan. Due to concerns of potential endocrine disruption, mothers were initially not randomized to toothpaste but could continue using their preferred product during pregnancy. At the first post-delivery home visit (~2 months postbirth), either triclosan-containing or triclosan-free toothpaste was provided according to assigned arm. Supplies were replenished every 4 months as needed during home visits.

In total, 154 mothers were enrolled and 136 remained enrolled after the infants were born. Household visits were conducted every 4 months to collect demographic and household information as well as stool and urine samples. Samples were stored at −80°C until processed. Automated weekly surveys on breastfeeding, diet, infant illness, including antibiotic use, were conducted, and infant medical records were referenced as available to account for antibiotic use around the time of sample collection. Of the 136 mother–baby pairs in STORK, 39 provided sufficient samples to be enrolled in this study. Of these 39 infants, 34 had medical record verification of

systemic antibiotic administration through the first year of life; 67.6% of infants in this study received systemic antibiotics (76.4% nTC, 60% TC), but administration did not occur within 1 month of sample collection in 95% of cases (104 of 109 infant samples). No antibiotic data were available for the mothers.

For gut microbiome analysis, we included all households with at least five of the expected six stool samples; 13 of the 39 households were missing one sample. From 13 households, one sample was missing: five were missing a sample from the mother (ID: 1002, 1084, 2360, 2443, 2584) and eight from the infant (ID: 1009, 2137, 2201, 2274, 2284, 2341, 2421, 2534).

### Urinary triclosan detection

To determine whether the randomization to TC-containing products was sufficient to increase TC exposure, urine samples were obtained from mothers and infants at the 6-month visit to measure urine triclosan levels. This time point was chosen because it followed randomization to all products, including toothpaste. Urinary triclosan levels (triclocarban was not assessed) were measured using liquid chromatography–mass spectrometry at the Stanford University Mass Spectrometry core facility. Urine samples (1 ml) were subjected to liquid–liquid extraction with ethyl acetate. Stable isotope-labeled triclosan (13C12, 99%, Cambridge Isotope Laboratory) served as the internal standard, and blank urine from subjects with no to minimal exposure to triclosan was used as sample matrix for calibration curve standards. The upper organic phase was collected and dried under a stream of nitrogen gas. Samples were reconstituted in 100 μl of 20% methanol and transferred to auto-sampler vials. The LC-MS/MS analysis was performed on a TSQ Vantage triple quadrupole mass spectrometer coupled with an Accela 1250 HPLC (Thermo Fisher Scientific). Injection volume was 10 μl. Reversed-phase separation was carried out on a Kinetex C18 column (50 × 2.1 mm ID, 2.6 μm particle size, Phenomenex). Mobile Phase A was water, and mobile phase B was methanol; flow rate was 350 μl/min. The gradient was as follows: 0 min (20% B), 2.5 min (98% B), 4 min (98% B), 4.5 min (20% B), and 6 min (20% B). The mass spectrometer was operated in negative APCI mode, with selected reaction monitoring (SRM).

Three SRM transitions were used for each triclosan and triclosan IS: 250.9 > 159.1, 187.0, 214.9, and 263.07 > 169.0, 197.9, 226.9, respectively. The calibration curve was linear from 1 to 40,000 fmol/μl, and the lower limit of quantitation (LLOQ) was around 10 fmol/μl of triclosan in extracted urine. Samples were measured in triplicate. All results were divided by 10 to account for concentration. Given the skewed distribution of triclosan levels, we used a nonparametric Mann–Whitney *U*-test to determine triclosan level differences between intervention arms.

### DNA extraction, 16S ribosomal DNA amplification, and amplicon sequencing

Samples were prepared and sequenced in two batches with approximately equal TC and nTC households per batch (29 households in batch 1, 10 households in batch 2) with identical methods. Samples were incubated for 10 min at 65°C before bead beating. One 20-min round of bead beating was performed at room temperature, and samples were mixed by inversion. DNA was isolated from stool

samples using the PowerSoil Isolation Kit (Mo Bio Laboratories, Inc., Carlsbad, CA, USA) per manufacturer's instructions. The protocol was modified to use approximately half the suggested weight (125 mg vs. 250 mg) of stool per sample. Region-specific primers, which included Illumina adapter sequences and 12-base barcodes on the reverse primer, were used to amplify the V4 region of the 16S ribosomal RNA gene. Failed reactions were rerun and amplicons were cleaned using UltraClean-htp 96-well PCR Clean-Up Kit (Mo Bio Laboratories, Inc.). Samples were quantified using Quant-iT dsDNA Assay Kit High Sensitivity (Thermo Fisher Scientific, Waltham, MA, USA) and measured on a FLEXstation II 384 microplate reader in the Stanford High-Throughput Bioscience Center. Amplicons were then combined in equimolar ratios, ethanol-precipitated, and gel-purified. Paired-end, 250-bp sequencing was performed on an Illumina MiSeq at the Stanford Functional Genomics Facility. A median of 47,143 (6,434–315,978) reads were sequenced per sample.

## Sequence processing and classification

We used a non-clustering method for 16S rRNA sequence classification using BaseSpace Application 16S Metagenomics v1.0 (Illumina, Inc.). Non-clustering implies amplicon reads are not grouped given a similarity score (typically 97–99%) to account for sequencing errors prior to classification. One caveat of clustering is that it reduces fine-scale variation that is biologically important. Briefly, the BaseSpace pipeline trims the 3′ ends of non-indexed reads when the quality score is < 15. High-quality reads were classified using a modified Ribosomal Database Project (RDP) Classifier (Wang *et al*, 2007) with a curated version of the Greengenes May 2013 reference taxonomy database. The original RDP classifier algorithm uses 8-base *k*-mers for classification; BaseSpace RDP uses 32-base *k*-mers, giving each *k*-mer more specificity for a given species. A curated version of the Greengenes May 2013 reference taxonomy filters those entries with 16S sequence length < 1,250 bp, more than 50 wobble bases, or those not classified at the genus or species level. The pipeline does not specifically check for chimeras; however, if the forward and reverse reads do not map to the same sequence in the reference database, they are excluded from classification.

## Prevalence taxonomy filtering

To filter rare (i.e., noisy) taxonomically classified reads, we calculated a prevalence threshold based on taxa found in at least seven samples. This threshold was chosen to include taxa that constitute a "core" microbiome, which suggests a taxon is persistent within at least two mothers throughout the study or a taxon during development is found in at least 20% of infants at one visit. This filtering resulted in the inclusion of 1,115 taxa from a pool of 1,892 taxa.

## Non-clustering classification methods comparison

We compared the BaseSpace RDP algorithm to another non-clustering algorithm, DADA2 (Callahan *et al*, 2016). For DADA2, 3′ ends of non-indexed reads with a quality score < 15 were removed using Trim Galore v. 0.4.1 (http://www.bioinformatics.babraham.ac.uk/projects/trim_galore/) to match BaseSpace RDP quality control. Then, the first 10 bases were trimmed from forward and reverse

reads (http://benjjneb.github.io/dada2/tutorial.html) following the developer's suggestions prior to dereplication, read merging, sample inference, and chimera checking steps using the DADA2 v. 1.1.1 package. Sequences were assigned taxonomy using the original RDP Classifier (https://rdp.cme.msu.edu/classifier/classifier.jsp) using the 16S rRNA training set 16. We followed the same filtering guidelines previously described, keeping taxa found in at least seven samples. This filtering resulted in the inclusion of 269 taxa from a pool of 4,772 taxa.

Median classification counts for the BaseSpace RDP Classifier and DADA2 can be found in Appendix Table S2. DADA2 has a median of 19,390 sequences retained per sample after filtering with 18 unique genera; BaseSpace RDP has a median of 47,135 sequences retained per sample with 110 unique genera identified. The large disparity of reads retained between the two methods can be explained by the sensitive sample inference algorithm to infer sequence uniqueness by DADA2, which leads to fewer reads per amplicon sequence considered a unique taxonomic unit. Our strict filtering of "core taxonomy" described in the main text penalizes taxa with fewer reads, leading to a large loss of taxa when compared to BaseSpace RDP. A PCoA was performed to compare the BaseSpace RDP and DADA2 taxonomic classifications on identifying variability between the samples. BaseSpace RDP and DADA2 methods are comparable on the first two principal components, separating infants and mothers despite fewer unique taxa in the DADA2 set compared to BaseSpace RDP (Appendix Fig S2). This similarity suggests variability in the study is driven by a low number of large effect taxa. Given the larger proportion of total and percent classified taxa at the genus level for BaseSpace RDP compared to DADA2, subsequent analyses proceeded with BaseSpace RDP classifications.

## Characterization of antibiotic resistance profiles

DNA archived from the previously described stool extraction was processed for shotgun DNA sequencing using the Nextera XT DNA Library Preparation Kit (Illumina Inc.) per manufacturer's instructions on a subset of 24 maternal samples from the 6-month visit. Paired-end, 101-bp sequencing was performed on an Illumina HiSeq 4000 at the Stanford Sequencing Service Center with an average of 21,660,032 (12,825,854–31,271,994) reads per sample.

Low-quality read ends with a Phred score < 20 were trimmed using TrimGalore v. 0.4.1 (http://www.bioinformatics.babraham. ac.uk/projects/trim_galore/), and PCR duplicates were removed using Super-Deduper v. 1.40 (http://dstreett.github.io/Super-Deduper/). High-quality reads were then aligned to the CARD (McArthur *et al*, 2013) using Burrows-Wheeler Aligner v. 0.7.10 (http://bio-bwa.sourceforge.net/). A median of 25,064 (8,200–72,464) reads were mapped to CARD per sample, with a median mapping percent of 0.11% for both intervention arms after adjusting for number of sequenced reads. A median of 5,476 (763–22,205) reads per sample mapped to genes implicated in resistance to triclosan (reviewed in Carey & McNamara, 2014). TC samples had a median of 7,043 reads mapped to antibiotic resistance genes (relative % adjusted for sequencing depth: 0.030%) compared for 5,330 reads (relative % adjusted for sequencing depth: 0.027%) for nTC samples. Euclidean distance was calculated between samples and then clustered with a hierarchical agglomeration method using base R functions.

## Statistical analyses

Analyses were performed in "R" v. 3.2.4 (http://www.R-project.org) with accompanying packages on non-rarified unique taxonomic classifications (McMurdie & Holmes, 2014). Principal coordinate analyses (PCoA) were performed using non-metric Bray–Curtis dissimilarity for combined analyses using "phyloseq" v. 1.14 (McMurdie & Holmes, 2013). Levels of triclosan in household products are intended to inhibit bacterial growth rather than kill bacteria (Suller & Russell, 2000). We hypothesized that the suppression of growth would alter taxa abundances rather than the presence/absence of taxa. Therefore, Bray–Curtis, a non-phylogeny-based method that takes abundance into account was chosen for PCoA. Sample distances for maternal-only analyses (Fig EV2) were calculated using the Canberra distance, which is best used for centroid-type patterns since maternal points on the overall PCA were centralized (Fig 1).

Alpha diversity measured by the Shannon diversity index and species richness measured by Chao1 was calculated using "vegan" v. 2.4 (Dixon, 2003). Nonparametric Mann–Whitney $U$-tests were used to determine statistical differences in diversity between intervention arms at each time point given the low sample sizes per comparison. To statistically test treatment effects on the homogeneity of microbial community composition, we performed permutational multivariate analysis of variance (PERMANOVA) analyses on distance metrics with "vegan" v. 2.4; 1,000 permutations were performed for mother and infant groups, stratified by the visits to account for infant development and maternal toothpaste introduction after the first visit as confounders. Differential taxa abundance and gene analyses were performed using "DESeq2" v. 1.10 (Love *et al*, 2014). A conservative 1% FDR-adjusted *P*-value threshold was selected for the treatment comparisons across mothers and infants pooling all three visits. This threshold is relaxed to 5% for comparisons at each time visit given lowered power from smaller sample size.

## Data availability

Raw 16S ribosomal RNA sequences for samples in this study are deposited in SRA under Bioproject accession PRJNA391974.

**Expanded View** for this article is available online.

## Acknowledgements

The authors thank Karolina Krasinska, the Vincent Coates Foundation Mass Spectrometry Laboratory, and the Stanford University Mass Spectrometry Laboratory (http://mass-spec.stanford.edu) for assistance developing and running the triclosan assays, and Susan Holmes and Eli Moss for helpful discussions regarding the analyses. This research was supported by the National Institutes of Health, Institute of Environmental Sciences (Grant no. R21 ES023371), by the Eunice Kennedy Shriver National Institute for Child Health and Human Development (R01 HD063142), and by a gift from Robert C. and Mary Ellen Waggoner. J.V.R. is funded by the National Science Foundation Graduate Research Fellowship (DGE-114747). Any opinion, findings, and conclusions or recommendations expressed in this material are those of the authors and do not necessarily reflect the views of the National Science Foundation. A.S.B. is funded by NCI K08 CA184420 and the Damon Runyon Cancer Research Foundation.

**The paper explained**

**Problem**

Triclosan and triclocarban (referred to as TCs) are the most common antimicrobials in household and personal care products. These antimicrobials were banned in the European Union in 2010, given concerns of endocrine disruption, induction of antibiotic resistance, and the disruption of aquatic communities. In 2016, the US Food and Drug Administration banned TCs and 17 other microbicides in some household and personal wash products. The extent to which TC exposure perturbs microbial communities in humans, particularly during infant development, was unknown.

**Results**

Our results suggest that TC exposure results in biologically plausible, though modest, effects in the gut microbiota of both adults (mothers) and infants. In mothers, we observed evidence of TC-induced taxon enrichment. These phylum level enrichments may be associated with broad antibiotic resistance and temporally correlate with introduction of TC-containing products not covered by the FDA ban.

**Impact**

Our study highlights the need for consumer safety testing for daily use antimicrobial products not subject to the FDA ban, and for further testing of the impact of these products on the human microbiome.

## Author contributions

JP and CL conducted the cohort study and nested intervention from which the specimens were collected and oversaw collection and coordination of all samples and metadata. TDH coordinated processing of stool and urine samples and extracted and amplified DNA from the stool samples. ET prepared the shotgun sequencing libraries. ASB and JVR designed the data analysis; JVR performed the computational and statistical analysis; JVR, ASB, and JP participated in data analysis and manuscript preparation; all authors edited the manuscript.

## Conflict of interest

The authors declare that they have no conflict of interest.

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
