## [Review Process File · EMBO Molecular Medicine]

Household triclosan and triclocarban effects on the infant and maternal microbiome

Jessica V. Ribado, Catherine Ley, Thomas D. Haggerty, Ekaterina Tkachenko, Ami S. Bhatt, Julie Parsonnet

Corresponding authors: Ami S. Bhatt and Julie Parsonnet, Stanford University

Review timeline:

Submission date:	10 April 2017
Editorial Decision:	05 May 2017
Revision received:	11 July 2017
Editorial Decision:	16 August 2017
Revision received:	22 August 2017
Editorial Decision:	24 August 2017
Revision received:	05 September 2017
Accepted:	11 September 2017

Transaction Report:

Editor: Céline Carret

1st Editorial Decision

05 May 2017

Thank you for the submission of your manuscript to EMBO Molecular Medicine. We have now heard back from the two referees whom we asked to evaluate your manuscript.

You will see from the comments below that both referees find the study interesting, while highlighting some shortcomings. While referee 1 is supportive and provides detailed suggestions to address these and improve the conclusiveness of the data, referee 2 is less so. Nevertheless as the main issues are shared, we would like to encourage you to address the referees' concerns in a major revision of your work. Please make sure to follow their advice and recommendations. We would not ask you to provide a validation in a different cohort as this is a randomized trial, we believe this to be significant, but strong claims should be moderated to better reflect the results.

Revised manuscripts should be submitted within three months of a request for revision; they will otherwise be treated as new submissions, except under exceptional circumstances in which a short extension is obtained from the editor.

Please note that EMBO Molecular Medicine policy encourages a single round of revision and that, as acceptance or rejection of the manuscript will depend on another round of review, your responses should be as complete as possible.

EMBO Molecular Medicine has a "scooping protection" policy, whereby similar findings that are published by others during review or revision are not a criterion for rejection. Should you decide to

submit a revised version, I do ask that you get in touch after three months if you have not completed it, to update us on the status.

I look forward to receiving your revised manuscript.

***** Reviewer's comments *****

Referee #1 (Comments on Novelty/Model System):

See below

Referee #1 (Remarks):

This is an exciting and extremely interesting study, well-designed and well-conducted. The approach and analysis are sound, and the writing is clear. The significance of this study is high, and it will bear an impact with respect to public health and future analysis. It was a pleasure to read and review this manuscript, and I applaud the team of investigators for their fine work.

There are a few factors and suggestions which, when considered and the manuscript revised, may lead to conclusions which are optimally supported by the data as presented.

1. Lack of infant gut microbiome variation with triclosan exposure: In Figure 1 the investigators show that triclosan levels, as measured by urinary secretion, differ by randomization groups for the mothers but not the infants at 6 months of exposure. Was this all true at 10 months?

This absence of cumulative dose in the infants is ultimately challenging for the investigators, since it renders the concern that the observed changes in the moms but not the infants microbiota may be largely dose related. This is emphasized in Figure 2, whereby infants show no triclosan exposure effect but moms do. In order to address the possibility that this is dose related, one option might be to compare taxonomic differences in the upper and lower tertiles of triclosan exposure for both mom and infant cohorts at each exposure time interval. This dose dependency measure could be similarly applied to the alpha diversity metrics presented in Figure 3.

On a more subtle note, the investigators comment on lines 103-105 that "Variations between infants were not driven by factors known to influence microbial colonization, such as delivery method, breast feeding, and pets in the household (Supplementary figure 4)." Although they do not provide any statistical estimate of clustering (such as PERMANOVA or Adonis), it appears that in their cohort there is no distinction by any of those factors in their cohort with the sole exception of formula only feeding (Figure S4). The investigators should make that clear, provide estimates of significance, and cite other work which similarly shows no impact by mode of delivery on the infant microbiome.

As a second minor comment, it may be worthwhile to look at several alpha diversity measures including Shannon and Chao1, as well as richness by Renyi.

2. Taxonomic variation with triclosan exposure: I am intrigued by the *B. fragilis* differences in the non-triclosan infant cohort, given its previous observations related to mode of delivery among some (but not all) other groups. I am curious if there was any colinearity with either mode of delivery or feeding with the *B. fragilis* infant variations projected in Figure 4. Based on the Cesarean delivery rate in their cohort, it would appear that since *B. fragilis* was the persistent most enriched species at 6 months in the nTC cohort, and that there was no effect of mode of delivery. This is worth highlighting.

3. Absence of observed variation in beta nor alpha diversity among infants from triclosan exposed households: The evidence presented shows no difference among infants, yet both the abstract and the summary state that one exists. These need to be clearly rewritten to reflect the data presented.

As noted above, without further dose dependency analysis we do not know if the absence of impact on the infant is a measure of diminished dose or rather a resilience of the developing microbiome. This is important to vet further.

4. Functional data: It would be nice to both see the evidence supporting the statement on lines 214-216 pertaining to antibiotic resistance genes as well as some measure of inferred functional metagenomics. Which pathways vary? Until presented, it is difficult to know if the investigators statements at several points pertaining to "potential harmful effects on host physiology" may actually be true, since neither host physiology nor bacterial functional pathways were formally examined.

Referee #2 (Comments on Novelty/Model System):

My ratings for novelty and impact are driven by existing literature that similarly supports the author's claims. Other studies (mSphere 2016) have reported modest affects on the microbiome although using random crossover study designs. Further impact to the medical field is seen as low largely because of the small sample size, lack of validation cohorts, and lack of mechanistic data that identifies how triclosan exposure through various routes alters the microbiome. Lastly, the authors' discussion of health implications and suggestions of harmful effects on host physiology are not appropriate.

Referee #2 (Remarks):

The authors report that triclosan exposure influences, to a very modest extent, the microbiome and that oral exposure (triclosan containing toothpaste) more significantly impact mothers compared to wash products. While technically sound, the results remain very preliminary and many of the statements provided by the authors are not supported by the data. First, the sample size is very small making it hard to assess any effects observed. It would seem prudent to conduct this study in an independent cohort to strengthen these claims. Second, the triclosan concentrations span several orders of magnitude but not clear dose-dependent effect can be found in the data. This is likely driven by the first point. Third, many of the claims in the discussion are not supported by the data. I am not sure statements regarding health implications or harm to host physiology are appropriate based only upon taxonomic profiling and metagenomics.

1st Revision - authors' response

11 July 2017

“Referee #1 (Remarks):

“This is an exciting and extremely interesting study, well-designed and well-conducted. The approach and analysis are sound, and the writing is clear. The significance of this study is high, and it will bear an impact with respect to public health and future analysis. It was a pleasure to read and review this manuscript, and I applaud the team of investigators for their fine work. There are a few factors and suggestions which, when considered and the manuscript revised, may lead to conclusions which are optimally supported by the data as presented.”

We sincerely appreciate Reviewer 1's enthusiasm for our work and specifically are thankful that he/she has noted that the study is "interesting ..., well-designed and well-conducted". We also are reassured that the reviewer felt the study was sound, written clearly and that the "significance of this study is high". We have sought to incorporate this helpful reviewer's suggestions and believe that in doing so, we have significantly improved our manuscript.

“1. Lack of infant gut microbiome variation with triclosan exposure: In Figure 1 the investigators show that triclosan levels, as measured by urinary secretion, differ by randomization groups for the mothers but not the infants at 6 months of exposure. Was this all true at 10 months? This absence of cumulative dose in the infants is ultimately challenging for the investigators, since it renders the

concern that the observed changes in the moms but not the infants microbiota may be largely dose related. This is emphasized in Figure 2, whereby infants show no triclosan exposure effect but moms do. In order to address the possibility that this is dose related, one option might be to compare taxonomic differences in the upper and lower tertiles of triclosan exposure for both mom and infant cohorts at each exposure time interval. This dose dependency measure could be similarly applied to the alpha diversity metrics presented in Figure 3.”

We thank the reviewer for this comment. The reviewer properly notes that triclosan levels in the urine are variable; this is attributed to triclosan’s relatively short half-life and the fact that spot triclosan level measurements depend on recent rather than sustained exposure. As such, it is likely that urinary triclosan measurements at 10 months are not an adequate measurement of cumulative dose, just as the 6 month levels are not. This, and the relatively high cost of triclosan measurement, is likely why the majority of research published on triclosan exposures reports spot triclosan levels at a single time point.

We agree that a measure of cumulative dose would be ideal. However, given the absence of any standard and accepted measurement of cumulative dose, we have taken the excellent suggestion provided by Reviewer 1. Specifically, we have divided the infant and mother cohorts into tertiles and as reviewer 1 suggests, “compare[d] taxonomic differences in the upper and lower tertiles of triclosan exposure effect”. Due to our limited power of detection caused by removing a third of our samples, we adjusted the threshold for statistically differential taxa from an FDR adjusted p-value of <0.05 to FDR adjusted p-value < 0.1, which is still fairly conservative. In so doing, we identified 10 differentially abundant taxa in upper and lower TC levels in mothers, including a Proteobacteria enrichment in higher TC levels. Infants had 16 differentially abundant taxa, most notably uncovering an enrichment of the Proteobacteria phylum in the higher TC levels and B. fragilis enrichment in lower TC levels. Stratification by urinary triclosan levels show enrichment trends consistent with the TC v. nTC comparison in mothers and uncover similar trends in the infants. These results have been added to the main text (lines 145). The full list of differentially abundant taxa for mothers and infants in the high and low urinary triclosan groups is reported in the appendix (Appendix Table S2).

*“We additionally compared differential taxa in the upper and lower tertiles of urinary TC levels at the 6 month visits, agnostic to treatment assignment. In the mothers, the taxa with the largest fold changes in the higher TC levels are members of the Proteobacteria phylum, and the infants with higher TC levels show a statistically significant enrichment of the Proteobacteria phylum and B. fragilis enrichment in lower TC levels (FDR adjusted p-value < 0.1, **Appendix Table S2**).”*

We also show that urinary triclosan levels do not linearly correlate with alpha [Shannon] diversity (line 126, Fig EV4).”

*“Microbial diversity did not correlate to urinary triclosan levels at 6 months (Infants: $R^2=0.025$, $p=0.40$; Mothers: $R^2=0.015$, $p=0.49$, **Fig EV4**). Species richness, measured by Chao1, is also not affected by TC exposure (p-values for 2, 6, and 10 months for infants: 0.61, 0.51, 1; mothers: 0.68, 0.49, 0.84).”*

Fig EV4: Microbial diversity does not correlate to urinary triclosan levels at 6 months for mothers or infants. Correlations were determined using a linear regression model (Shannon diversity ~ Urinary triclosan levels) on mother and infant samples separately.

“On a more subtle note, the investigators comment on lines 103-105 that “Variations between infants were not driven by factors known to influence microbial colonization, such as delivery method, breast feeding, and pets in the household (Supplementary figure 4).” Although they do not provide any statistical estimate of clustering (such as PERMANOVA or Adonis), it appears that in their cohort there is no distinction by any of those factors in their cohort with the sole exception of formula only feeding (Figure S4). The investigators should make that clear, provide estimates of significance, and cite other work which similarly shows no impact by mode of delivery on the infant microbiome.”

We thank the reviewer for drawing attention to this oversight. Specifically, we have added statistical estimates for clustering for additional factors known to affect microbiome composition in infants, such as delivery method and breast feeding, not clearly shown by principle coordinate analyses (line 111, Fig EV3). Some factors, while predicting minor variance, are significant using a PERMANOVA model. We have replaced former text with the sentence:

“The low contribution to variance of factors known to influence microbial colonization in infants by 2 months of age, such as delivery method ($R^2=0.068$, $p=0.011$) and breast feeding ($R^2=0.13$, $p=0.041$ in the household (Fig EV3) is consistent with previously reported data (Chu et al., 2017).”

“As a second minor comment, it may be worthwhile to look at several alpha diversity measures including Shannon and Chao1, as well as richness by Renyi.”

We appreciate this suggestion to improve the robustness of our analysis and have reported the results of applying the Chao1 richness estimator, and showed that richness estimates are not different between treatment arms for infants and mothers (line 128).

*“Relative abundances of *B. fragilis* at this visit were not associated with delivery method (Mann Whitney U-test $p=0.85$) or breast feeding (ANOVA $p=0.52$) as previously reported (Martin et al., 2016).”*

“2. Taxonomic variation with triclosan exposure: I am intrigued by the *B. fragilis* differences in the non-triclosan infant cohort, given its previous observations related to mode of delivery among some (but not all) other groups. I am curious if there was any collinearity with either mode of delivery or feeding with the *B. fragilis* infant variations projected in Figure 4. Based on the Cesarean delivery rate in their cohort, it would appear that since *B. fragilis* was the persistent most enriched species at

6 months in the nTC cohort, and that there was no effect of mode of delivery. This is worth highlighting.”

The reviewer notes important findings in recent literature regarding B. fragilis and delivery method. We followed-up the conflicting literature on B. fragilis colonization with mode of delivery and breast feeding. We performed a Mann Whitney U test for Caesarean v. vaginal birth and an ANOVA for the 4 feeding categories of feeding on the relative abundance of B. fragilis in infant samples at 6 months. We found that there is no colinerarity with mode of delivery or breastfeeding with the relative abundance of B. fragilis, and have added this point to the main text (line 142).

“Relative abundances of B. fragilis at this visit were not associated with delivery method (Mann Whitney U-test $p=0.85$) or breast feeding (ANOVA $p=0.52$) as previously reported (Martin et al., 2016).”

“3. Absence of observed variation in beta nor alpha diversity among infants from triclosan exposed households: The evidence presented shows no difference among infants, yet both the abstract and the summary state that one exists. These need to be clearly rewritten to reflect the data presented. As noted above, without further dose dependency analysis we do not know if the absence of impact on the infant is a measure of diminished dose or rather a resilience of the developing microbiome. This is important to vet further.”

We regret that our original statements about the lack of difference among the infants was unclear, and have now clarified our text to clearly state diversity is not affected, but minor taxonomic enrichments or Proteobacteria are seem in TC mothers.

However, we do agree that without cumulative dose analyses, the importance of diminished dose or resilience of the developing microbiome should be highlighted as possible reasons for the absence of impact on the infant microbiome. We have included this alternative confounder in our discussion (line 209).

“Given these perturbations were present in both arms of the study, it is possible the dearth of taxonomic differences in the infant samples emphasize the resiliency of the developing microbiome. However, the trend of Proteobacteria enrichment in infants with higher urinary triclosan levels suggest the small microbiota disruptions observed in the study were TC dose dependent.”

“4. Functional data: It would be nice to both see the evidence supporting the statement on lines 214-216 pertaining to antibiotic resistance genes as well as some measure of inferred functional metagenomics. Which pathways vary? Until presented, it is difficult to know if the investigators statements at several points pertaining to "potential harmful effects on host physiology" may actually be true, since neither host physiology nor bacterial functional pathways were formally examined.”

We agree that the provided analysis, which focuses on the presence of antibiotic resistance genes, does not address overall functional capacity of the microbial communities. Given the relatively small number of individuals studied, and the very large number of measurements that would be performed when carrying out a functional analysis of gene presence, we concluded that a complete functional analysis would be statistically underpowered and subject to potential overinterpretation. As such, we did not intend for the whole shotgun sequencing to be a full functional approximation of gut metagenomes in the mothers. Instead, we focused on our hypothesis that the expansion of broadly antibiotic resistant Proteobacteria may suggest a higher proportion of antibiotic resistant genes in the stool of TC mothers. We described the procedure of aligning shotgun reads to the antibiotic database for this focused analysis in line 152. We have removed “potential harmful effects on host physiology”, as both reviewers 1 and 2 point out that this is an overinterpretation of the presented data. Potential antibiotic resistant strains identified in our intestinal study may not be pathogenic themselves, but the possibility for horizontal gene transfer with pathogenic strains between people and the environment, may severely exacerbate problems with population resistance. We clarified our concerns of the potential spread of antibiotic resistance highlighted by the Proteobacteria expansion and the enrichment of a beta-lactamase antibiotic resistance gene in lines 129 and 135.

“Potential antibiotic resistant strains identified in this study may not be pathogenic, but the possibility for horizontal gene transfer with pathogenic strains between people and the environment may severely exacerbate problems with population and clinical resistance.” “Although we only identified one significantly enriched antibiotic-resistance gene in the TCexposed mothers, point mutations in the candidate, CfxA6, have led to antibiotic resistance in the gut following antibiotic exposure (Raymond et al., 2016).”

Referee #2 (Comments on Novelty/Model System):

“My ratings for novelty and impact are driven by existing literature that similarly support the authors claims. Other studies (mSphere 2016) have reported modest affects on the microbiome although using random crossover study designs. Further impact to the medical field is seen as low largely because of the small sample size, lack of validation cohorts, and lack of mechanistic data that identifies how triclosan exposure through various routes alters the microbiome. Lastly, the authors discussion of health implications and suggestions of harmful effects on host physiology are not appropriate.”

We appreciate that the reviewer has identified the one other randomized study of triclosan in the literature. As noted, this is not the first randomized study to evaluate the impact of triclosan on adults While limited data does exist regarding the impact of oral triclosan exposure on the gut microbiome (mSphere 2016), to our knowledge this is the first study of this kind of infants, and spans a longer duration (>4 months) than previously reported in adults. A fundamental limitation of randomized-prospective trials in humans is that they are not well-suited to evaluate questions of mechanism. Despite this, we appreciate Reviewer 2’s comments on the important of evaluating mechanism and thus we attempted to evaluate mechanism by which triclosan may impact the microbiome. Specifically, we followed up on the suggestions of antibiotic resistance with the Proteobacteria enrichment by quantifying known triclosan antibiotic resistance genes (Fig EV5) but were hindered by a low sequencing coverage of these genes.

As Reviewer 2 and the editor can likely appreciate, due to the significant cost and effort involved in carrying out a randomized-controlled study, typically, one does not report a validation randomized-controlled study with the original report of a randomized-controlled study. Additionally, given the FDA bans on some products used in this study, conducting a second randomized trial will not be possible.

However, we did attempt to evaluate the reproducibility of our findings in previously published cohort. To evaluate the validity of Proteobacteria enrichment in our STORK cohort, we reanalyzed previously published data from a randomized, crossover study of TC household products (Poole et al. 2016). Briefly, 14 individuals were assigned to either a TC or nTC arm for 4 months, then switched to using nTC or TC products for an additional 4 months respectively. We downloaded raw sequencing files from the European Nucleotide Accession (ERA556305) and classified reads using similar methods as described in our study. This data, despite filtering, was sparser than our randomization trial, meaning that many taxa were not present across samples in the same phase. This data structure violates the DESeq2 model, which assumes that most counts are non-zero. To overcome sparsity in the Poole data, we used a more conservative, non-parametric Kruskal-Wallis sum-rank test to detect differentially abundant taxa combined with linear discriminant analysis to estimate the effect size of each taxon using LEfSe (<https://huttenhower.sph.harvard.edu/galaxy/>). We also ran LEfSe on the maternal samples at 6 months between nTC and TC arms for comparable results. Most species identified to be enriched in TC samples did not overlap across studies, but we did observe an overlap of the Klebsiella genus in both the Poole and our study at the 6-month visit. We cannot disentangle differences in sample collection, preparation, and gender disparities to explain low taxonomic overlap within and between studies, which would be required to provide a fair validation. Thus, given the high risk for overinterpretation of this secondary analysis of a previously published dataset, we have not included this analysis in our current manuscript.

We acknowledge that the n of this study is small, but it is the largest RCT of triclosan in the literature, to date. Additionally, it is interesting to note that despite the small n, we have performed careful statistical analyses and were powered to identify statistically significant differences between the triclosan-exposed and unexposed groups.

Finally, we sincerely appreciate Reviewer 2 noting some wording in our original text that invites over interpretation. We have taken reviewer 2's suggestion and removed "potential harmful effects on host physiology" since we were unclear, addressed in our response to reviewer 1, #4.

Referee #2 (Remarks):

"The authors report that triclosan exposure influences, to a very modest extent, the microbiome and that oral exposure (triclosan containing toothpaste) more significantly impact mothers compared to wash products. While technically sound, the results remain very preliminary and many of the statements provided by the authors are not supported by the data. First, the sample size is very small making it hard to assess any effects observed. It would seem prudent to conduct this study in an independent cohort to strengthen these claims. Second, the triclosan concentrations span several orders of magnitude but not clear dose-dependent effect can be found in the data. This is likely driven by the first point. Third, many of the claims in the discussion are not supported by the data. I am not sure statements regarding health implications or harm to host physiology are appropriate based only upon taxonomic profiling and metagenomics."

We appreciate Reviewer 2's assertion that our work is "technically sound" and have sought to clarify statements that may appear to be "not supported by the data". Specifically, we have removed the statements related to "potential[ly] harmful effects on host physiology" as noted above.

We agree that the sample size is relatively small, and as noted in the previous section would like to point out that this is the largest prospective RCT of triclosan exposure that has been carried out, to date. Furthermore, we are still powered to detect minor differences that have implications of antibiotic resistance. In response to the second comment, "no[t] clear dosedependent effect can be found in the data" – we have now addressed this shortcoming.

Specifically, following the suggestion of reviewer 1, differential abundance analyses by the dose tertiles in the mothers and infants at the 6-month visit are a proxy for dose-dependent TC exposure effects. We found that trends of Proteobacteria enrichment in TC households hold. Finally, in response to the reviewer's third point regarding "statements regarding health implications", as noted above, we have removed the statements related to "potential[ly] harmful effects on host physiology".

2nd Editorial Decision

16 August 2017

Thank you for the submission of your revised manuscript to EMBO Molecular Medicine. We have now received the enclosed report from the referee who was asked to re-assess it. As you will see, unfortunately the reviewer remains unsupportive. Before to move forward, we would really appreciate it if you could write a rebuttal to the referee's comments detailing your responses as much as possible and discussing the commented limitations to the best of your abilities, including within the paper. We would then editorially evaluate your responses and make a final decision then.

Please submit your revised manuscript as soon as possible for further evaluation. Thank you for your understanding.

***** Reviewer's comments *****

Referee #2 (Comments on Novelty/Model System):

The results are preliminary and not convincing to demonstrate that indeed triclosan promotes adverse effects on the gut microbiome. Other studies have similarly concluded that triclosan does not promote changes. For these reasons I have judged the novelty and medical impact to be low.

Referee #2 (Remarks):

I appreciate the authors attempt to meaningfully address the criticisms and agree that some of the important points have been addressed. However despite these efforts and while the manuscript and

its ideas are intriguing, the results and their interpretation remain preliminary. The sample size remains small and the notion that this is the largest study conducted to date is misleading since others have been small as well. Additionally, the authors focus on Proteobacteria, a major phylum of bacteria that indeed contains both non- and pathogenic organisms; however interpretation of the data using 16S rRNA sequencing and the pilot metagenomics analysis remains speculative as they cannot unequivocally identify the species of Proteobacteria or conclude that the observed changes (in this case very small even with adjusted FDRs) contribute to antibiotic resistance. Perhaps triclosan promotes more beneficial strains of Proteobacteria and this explains the lack of increased antibiotic resistance strains? Substantially more investigation is required to better appreciate the taxonomic changes reported. Another important issue is that Proteobacteria changes are often observed with antibiotic use as acknowledged by the authors. However, suggesting that antibiotic use overwhelmed any change with triclosan is premature as an equally plausible conclusion is that triclosan exposure in infants is not associated with any significant or durable changes. A major concern surrounds the issue of dose-response which is likely reflected by differences in time of collection, the short half life of triclosan, and the small sample size to discern a significant relationship (not trend). Other considerations include that while triclosan exposure does not impact the gut microbiome, perhaps it more substantially impacts the skin microbiome. This might be even more relevant for wash products as breastfeeding results in substantial skin-to-skin contact.

2nd Revision - authors' response

22 August 2017

Referee #2 (Comments on Novelty/Model System):

“The results are preliminary and not convincing to demonstrate that indeed triclosan promotes adverse effects on the gut microbiome. Other studies have similarly concluded that triclosan does not promote changes. For these reasons I have judged the novelty and medical impact to be low.”

We appreciate that the reviewer has noted previous studies have been conducted on triclosan and the microbiome in the literature. While limited data does exist regarding the impact of oral triclosan exposure on the gut microbiome (mSphere 2016), to our knowledge this is the first study of this kind of infants, and this spans a longer duration than the previous, noted study (1 year vs. 4 months). Furthermore, in this study, we identify new results – specifically, we see minor taxonomic differences in the TC exposed vs. unexposed arms in our study while the previously published reports in adults do not show a Proteobacteria enrichment. We have been careful to note that these differences are not strictly “adverse”, as we do not have long term health data that would be required to evaluate a health endpoint. These data provide a new perspective on TC exposure and the gut microbiome in adults that may have been undetectable due to smaller sample size in previously published studies. As would be expected, larger studies, such as ours, have greater power to detect smaller differences. It is important to note that since triclosan has been removed by the US FDA from wash products and was previously removed by the European authorities, it is unlikely we will see a study with more detailed information on this topic.

Referee #2 (Remarks):

“I appreciate the authors attempt to meaningfully address the criticisms and agree that some of the important points have been addressed. However despite these efforts and while the manuscript and its ideas are intriguing, the results and their interpretation remain preliminary. The sample size remains small and the notion that this is the largest study conducted to date is misleading since others have been small as well.”

We acknowledge that the n of this study is small. It is, however, the largest RCT of triclosan in the literature, to date. Our sample size of 78 individuals (39 infants and 39 adults) is appreciably larger than the previously published RCT of triclosan with 14 adults. Furthermore, we carried out the intervention for a full year, which results in a prolonged exposure. Despite the small n, we have performed careful statistical analyses and were powered to identify statistically significant differences between the triclosan-exposed and unexposed groups. Our results reflect statistically significant taxonomic difference observed in animal models exposed to triclosan which is described in the manuscript, supporting the biological plausibility of our results.

Line 191: *“The finding that Proteobacteria were enriched in the gut microbiota is consistent with observations in fish following triclosan exposure (Narowe et al., 2015).”*

“Additionally, the authors focus on Proteobacteria, a major phylum of bacteria that indeed contains both non- and pathogenic organisms; however interpretation of the data using 16S rRNA sequencing and the pilot metagenomics analysis remains speculative as they cannot unequivocally identify the species of Proteobacteria or conclude that the observed changes (in this case very small even with adjusted FDRs) contribute to antibiotic resistance. Perhaps triclosan promotes more beneficial strains of Proteobacteria and this explains the lack of increased antibiotic resistance strains? Substantially more investigation is required to better appreciate the taxonomic changes reported.”

We appreciate Reviewer 2’s concern for highlighting non-pathogenic commensals in the Proteobacteria phylum. We chose not to focus on the taxonomic enrichments and documented pathogenicity at lower taxonomic levels due to a risk for over-interpretation. This is particularly important as some enriched taxa cannot be classified down to the genus or species level. The current reference databases tend to favor more specific classification (down to genus, species level) of pathogens as opposed to commensals, as pathogens are over-represented in reference databases, and many non-pathogenic commensal species have yet to be sequenced and deposited in these databases. To clarify the Proteobacteria link and antibiotic gene resistance in the gut microbiome, we added a qualifying transition to our pilot metagenomics study and highlighted the difficulty of accessing clinical or public health significance to the phylum as a whole.

Line 156: *“Proteobacteria enrichment has been associated, but is not directly correlated, with increased antibiotic resistance genes in the human gut microbiota, independent of antibiotic exposure (Bengtsson-Palme et al., 2015). This suggests Proteobacteria species may not harbor antibiotic resistance genes in the gut microbiome, but expansion may serve as marker for increased antibiotic resistance in a community.”*

Line 231: *“The Proteobacteria phylum contains both pathogenic and nonpathogenic species, which makes it difficult to ascribe a clinical or public health relevance to our results.”*

We previously clarified that antibiotic resistance does not imply pathogenicity.

Line 239: *“Potential antibiotic resistant strains identified in this study may not be pathogenic, but the possibility for horizontal gene transfer with pathogenic strains between people and the environment may severely exacerbate problems with population and clinical resistance. “*

We acknowledge that our pilot study, which attempted to evaluate the differential abundance of antibiotic resistant genes in TC exposed vs. unexposed individuals, suffers from limited power. This is likely due to the fact that a low proportion of reads mapping to antibiotic resistance genes in our methods. Antibiotic resistance characterization using orthogonal methods is certainly called for in future studies to better understand TC effects on gut antibiotic resistance.

Line 165: *“Unsupervised clustering by triclosan resistance gene counts was not sufficient to cluster maternal samples by intervention arm, and this may have been impacted by the limited power to detect differences between groups due to low coverage.”*

Line 245: *“Although we only identified one significantly enriched antibiotic-resistance gene in the TC-exposed mothers, point mutations in the candidate, CfxA6, have led to antibiotic resistance in the gut following antibiotic exposure (Raymond et al., 2016). Future in vitro and potentially in vivo studies will be required to more thoroughly characterize the impact of TCs on antibiotic resistance in the gut microbiota.”*

We also previously addressed other associations of Proteobacteria enrichments with human health that are not antibiotic resistance related, and we agree that further investigation on these shifts is necessary to understand the relationship. Such studies are likely outside the scope of this study.

Line 230: *“While a selective expansion of Proteobacteria is not known to cause disease, Proteobacteria expansion has been proposed as a potential diagnostic signature of dysbiosis linked to diabetes, colitis, and malnutrition (Shin et al., 2015). Future studies may illuminate the impact of these shifts on health-related outcomes.”*

“Another important issue is that Proteobacteria changes are often observed with antibiotic use as acknowledged by the authors. However, suggesting that antibiotic use overwhelmed any change with triclosan is premature as an equally plausible conclusion is that triclosan exposure in infants is not associated with any significant or durable changes. A major concern surrounds the issue of dose-response which is likely reflected by differences in time of collection, the short half life of triclosan, and the small sample size to discern a significant relationship (not trend).”

We appreciate reviewer 2's comment on the potential antibiotic confounder and the suggestion to improve the clarity of this section. We included antibiotic administration data to highlight that known disruptors of the gut microbiome that are also associated with Proteobacteria enrichment are not seen in only one arm. We do not state their administration overwhelms the lack of taxonomic differences as suggested, and support this idea with the urinary level tertile analyses. Proteobacteria enrichment in infants with higher urinary triclosan levels is statistically significant, as highlighted in our results.

Line 152: “Infants with higher triclosan levels showed a statistically significant enrichment of the Proteobacteria phylum in their stools samples; conversely, with lower triclosan levels, stool samples were enriched for *B. fragilis* (FDR adjusted p-value < 0.1, **Appendix Table S2**).”

We changed “trend” to “statistically significant” to improve clarity.

Line 219: “However, the statistically significant Proteobacteria enrichment in infants with higher urinary triclosan levels suggest the small microbiota disruptions observed in the study were TC dose dependent.”

We acknowledge the concern surrounding variations in urinary triclosan detection. We recognize product usage varies between households and within households depending on the day which can alter the amount of triclosan measured in the urine. While the half-life of triclosan is approximately 24 hours, baseline levels are achieved after 8 days of single oral exposure (Calafat et al. 2008, doi: [10.1289/ehp.10768](https://doi.org/10.1289/ehp.10768)). Mothers in the TC arm are consistently using triclosan-containing toothpaste throughout the study within this time frame. Given that strong association of Proteobacteria enrichment in the mothers following toothpaste randomization, we feel confident that Proteobacteria enrichment is also seen in the upper urinary triclosan level tertile in infants is likely driven by changes in exposure levels.

“Other considerations include that while triclosan exposure does not impact the gut microbiome, perhaps it more substantially impacts the skin microbiome. This might be even more relevant for wash products as breastfeeding results in substantial skin to skin contact.”

We agree that the evaluating the impact of TC on the skin microbiome would be a very exciting follow-up to this study given the high exposure to TC containing products both by skin-to-skin contact and bathing in TC containing products, as mentioned in the discussion.

Line 252: “Given the high prevalence of TC exposure on the skin in this study, it will be interesting to study the impact of these wash products on the skin microbiota and related health outcomes.”

We are hoping to assess this possibility using archived skin swabs.

3rd Editorial Decision

24 August 2017

Thank you for the submission of your revised manuscript to EMBO Molecular Medicine and providing the rebuttal letter. We have now evaluated your responses to referee 2 and are satisfied with your answers and text inclusions. I am pleased to inform you that we will be able to accept your manuscript pending the following editorial amendments:

Main article:

-in legends, please add exact n= and exact p= values, not a range. Some people found that to keep the figures clear, providing a supplemental table with all exact p-values was preferable. You are

welcome to do this if you want to.

Please submit your revised manuscript within two weeks. I look forward to seeing a revised form of your manuscript as soon as possible.

3rd Revision - authors' response

05 September 2017

Authors made requested editorial changes.

Corresponding Author Name: Ami Bhatt
Journal Submitted to: EMBO Molecular Medicine
Manuscript Number: